# LEMUR: LEVERAGING VISION-LANGUAGE MODELS FOR FINE-GRAINED MULTIMODAL RETRIEVAL

## ABSTRACT

Fine-grained multimodal retrieval is crucial for many real-world applications. For example, E-commerce product search demands retrieving the product with the most relevant image and description based on specific regions of the query image. However, existing CLIP-based or VLM-based retrieval methods primarily focus on image-level tasks and struggle with region-level applications. In this work, we present LEMUR, a VLM-based fine-grained retrieval framework that enhances the regional representations without compromising its image-level retrieval performance. At its core, LEMUR incorporates a Region-Aware Encoder that extracts detailed features from query regions to complement the global image representation. To further enhance fine-grained retrieval capability, we integrate detailed localized captioning and regional contrastive learning tasks, which strengthen the model's fine-grained understanding and representation. In addition, considering the limitations of existing benchmarks, such as the absence of region-level contrastive pairs and the limited diversity of evaluation tasks, we introduce the FGMB benchmark. It comprises 225k contrastive pairs, covering two metatasks and four multimodal retrieval scenarios. Extensive experiments validate the effectiveness of our approach. LEMUR generally outperforms strong baselines in zero-shot settings. Further training with regional contrastive learning leads to an average improvement of 20% in fine-grained retrieval performance, while achieving comparable or better results in image-level retrieval tasks. The code and data will be released to facilitate future research.

## 1 INTRODUCTION

In recent years, multimodal information retrieval has been widely applied in diverse real-world scenarios, such as E-commerce product search (Jin et al., 2023; Dong et al., 2022), social media content recommendation (Yu et al., 2025a), medical visual question answering (Zhang et al., 2024b), and content generation (Yasunaga et al., 2022). Existing studies largely fall into two categories. The first is the CLIP-based method, such as Radford et al. (2021); Zhai et al. (2023); Li et al. (2022), which performs strongly on cross-modal tasks (e.g., image–text). The second is the VLM-based method (Jiang et al., 2024b;a), which extends the large language model's embedding capability to fused-modal retrieval (e.g., image–text-to-image). However, the current research community mainly focuses on image-level retrieval, while ignoring region-level fine-grained retrieval, which is critical in real-world applications. For example, a shopper might upload a street photo and select the bounding box around a passerby's coat, prompting the system to retrieve similar products based on region-specific cues like shape, color, and brand.

Some attempts have been made to address the fine-grained retrieval problem. Specifically, Xie et al. (2025); Jing et al. (2024) extend the CLIP-based methods by using RoIAlign (Abdulla, 2017) to get regional representation, and further enhance it through regional contrastive learning. However, their fine-grained embedding is only aligned with short and simple captions, which is ineffective in representing complex image regions. Meanwhile, they also struggle to generate unified embeddings for fused-modal queries, requiring specialized modifications as in Wei et al. (2024). Although VLM-based methods exhibit strong ability for fused-modal tasks, their potential for fine-grained retrieval remains underexplored. Some naive prompting strategies (Yang et al., 2023; Yu et al., 2025b) can repurpose them for region-level tasks. Yet, these models neither explicitly extract fine-grained features nor employ regional contrastive learning for better visual-semantic local alignment.

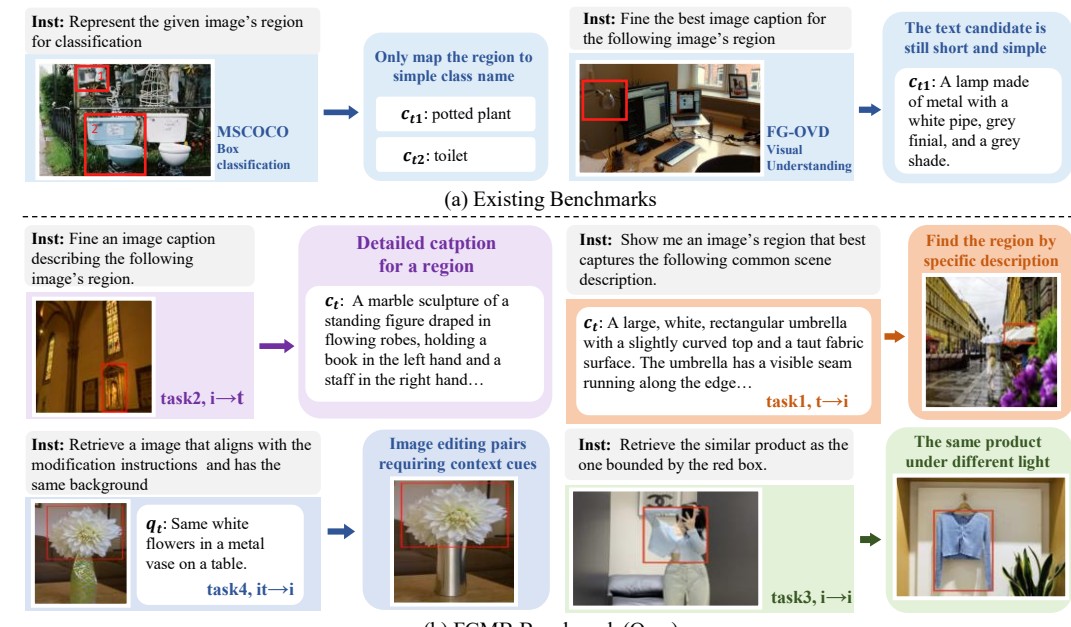

Figure 1: Examples of FGMB and existing benchmarks. Compared with FG-OVD and COCO, FGMB provides more complex query–candidate pairs and spans a wider range of modalities.

In this work, we introduce LEMUR, a VLM-based framework for fine-grained multimodal retrieval. At its core, LEMUR integrates a Region-Aware Encoder (RAE) alongside the native vision encoder (denoted as context encoder, CE) to explicitly capture detailed regional features. To enrich the regional representations with contextual cues, the RAE utilizes cross-attention with the image-level features from CE. The RAE is organized into a layer-wise paradigm with CE, allowing it to gather information at different semantic levels from all ViT layers. In addition, to prevent RAE's features from disrupting the VLM's global-level visual-semantic alignment, they are fed into the LLM via a separate projector. This decouple design preserves the image-level retrieval performance after training on fine-grained datasets. Furthermore, to equip LEMUR with better retrieval ability, we adopt a three-stage training strategy, including (1) RAE pretraining, which employs the Detailed Localized Captioning task (Lian et al., 2025) under the next-token prediction paradigm to encourage the model to learn highly detailed regional representations; (2) Language-only contrastive learning, which effectively converts the language generation capability of LLMs into embedding capability; and (3) Regional contrastive learning, which largely improves the fine-grained retrieval performance.

Additionally, to support training with regional contrastive learning and provide a comprehensive evaluation framework, we establish a new benchmark named FGMB. This benchmark covers two metatasks and four multimodal retrieval scenarios (Zhang et al., 2024c), targeting the challenge of retrieving the most relevant image or text given a query that specifies a region of interest. FGMB contains 225k region-level contrastive pairs, primarily derived from automatically annotated open-source web images (Byeon et al., 2022; Kirillov et al., 2023a), image editing datasets (Awal et al., 2024; Jiao et al., 2024), and further enriched with a manually curated and privacy-sanitized subset from an image search database from a popular E-commerce platform[1]. All benchmark data will be released to support future research.

Extensive experiments validate the effectiveness of our approach and dataset. In the zero-shot setting, LEMUR directly combines the pretrained RAE with existing image-level VLM embedding models. Without any additional region-level contrastive training, LEMUR can surpass strong baselines in fine-grained retrieval tasks. Moreover, fine-tuning on the FGMB train set further enhances performance, yielding up to a 30% improvement in region-level retrieval accuracy and providing gains for image-level retrieval tasks as well. In summary, our main contributions are as follows:

---

[1]We are authorized by the company to obtain and share the data. The dataset is released under CC BYNC-SA 4.0 license and can be used for noncommercial purposes.

- We propose LEMUR, a VLM incorporating a Region-Aware Encoder to process region-level prompts and produce fine-grained visual tokens. We further design a three-stage training pipeline to strengthen LEMUR's fine-grained understanding and embedding ability. LEMUR achieves state-of-the-art performance in the zero-shot setting, and its effectiveness can be further boosted through region-level contrastive training, while also benefiting image-level retrieval.

- We construct FGMB, the first fine-grained multimodal retrieval benchmark comprising 225k high-quality contrastive pairs, organized into two meta-tasks and four representative retrieval scenarios. It provides data for both training and evaluation, enabling comprehensive evaluation and facilitating deployment of fine-grained multimodal retrieval methods.

## 2 RELATED WORKS

**MLLMs for Multimodal Embedding.** Recent advances in multimodal large language models (MLLMs) have demonstrated remarkable performance in embedding learning. Early efforts, such as E5 (Wang et al., 2022) and NV-Embed (Lee et al., 2024), adapted the generative capabilities of large language models to text retrieval tasks. Building upon this, LamRA (Liu et al., 2025), Vlm2vec (Jiang et al., 2024b), E5-V (Jiang et al., 2024a), and MM-Embed (Lin et al., 2024) extended the paradigm from text retrieval to multimodal retrieval. More recently, MME5 (Chen et al., 2025b) leveraged synthetic data to further improve multimodal retrieval performance. MoCa (Chen et al., 2025a) transforms the causal language models into bidirectional encoders and thereby significantly enhances the representational capacity. Compared to classical retrieval approaches, these MLLM-based methods exhibit stronger generalization and representation abilities.

**Fine-Grained Representation Learning.** A growing line of work has sought to align the representations of image and text on the fine-grained level. RegionCLIP (Zhong et al., 2022), FGCLIP (Xie et al., 2025), and FineCLIP (Jing et al., 2024) employ RoIAlign to extract regional features and align them with fine-grained descriptions. LongCLIP (Zhang et al., 2024a) and DreamCLIP (Zheng et al., 2024) leverage long captions to elicit more detailed semantic grounding between images and text. In parallel, FLAIR (Xiao et al., 2025) adopts an extra language condition to derive the image representation of the target region. However, they often encounter challenges in multimodal retrieval due to the dual-encoder architecture and the simplistic processing of regional visual features.

**Multimodal Information Retrieval Benchmarks.** Representative image-level benchmarks include MMEB (Jiang et al., 2024b), M-BEIR (Wei et al., 2024), and MMEB-V2 (Meng et al., 2025), which cover a wide range of fused-modal tasks such as image-text-to-image retrieval. In contrast, fine-grained region-level benchmarks remain limited. The most widely adopted evaluation is to reformulate the MSCOCO box classification task (Lin et al., 2014) as a retrieval problem. In addition, FG-OVD (Bianchi et al., 2024) requires models to identify the most relevant description among multiple confusable candidates for a given region. However, these benchmarks are confined to image–text retrieval with relatively simple text, and therefore cannot adequately assess fine-grained retrieval performance in fused-modal settings.

## 3 METHODS

In this section, we first formulate the fine-grained multimodal information retrieval task in Section 3.1. Then we elaborate on LEMUR in Section 3.2, including the architecture of the Region-Aware Encoder and the way to handle region-level prompts effectively. Finally, in Section 3.3, we describe the three-stage training strategy designed to boost Lemur's retrieval performance.

### 3.1 TASK FORMULATION

Fine-grained multimodal retrieval aims to find the most relevant image or text at the region level. Formally, we define the query set as $Q = \{(q_1, r_{q_1}), (q_2, r_{q_2}), \ldots, (q_M, r_{q_M})\}$, where each query $q_k$ may consist of an image, a text description, or a combination of both. If the query includes an image, it is associated with a region of interest $r_{q_k}$; otherwise, the $r_{q_k}$ is empty. We denote by the operator $q \otimes r_q$ the process of constructing a region-specific query, where the image part of $q$ is cropped to the ROI $r_q$ and the text part remains unchanged. Similarly, we define the candidate set as $C = \{(c_1, r_{c_1}), (c_2, r_{c_2}), \ldots, (c_N, r_{c_N})\}$, where each $c_k$ comprises images, text, or interleaved

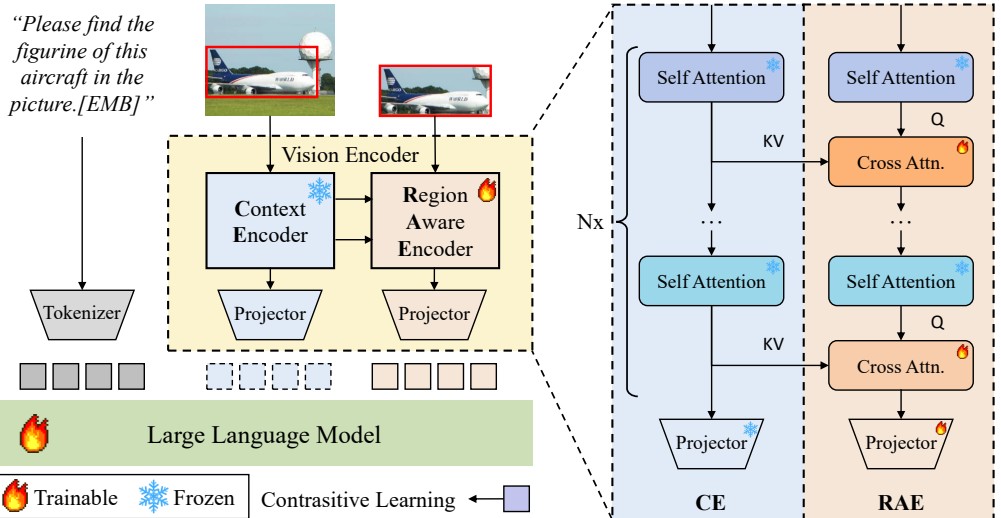

Figure 2: The overview of the LEMUR framework. The image and its region of interest are processed by the native visual encoder (CE) and the RAE, respectively. Each RAE layer applies cross-attention with CE's corresponding layer to extract region-specific signals at matching semantic levels. The attentions blocks in the same color share weights. CE tokens in the dashed box are optional to be used jointly with RAE tokens. A large language model integrates the multimodal tokens and produces a final embedding for retrieval.

formats. Note that as this study focuses on multimodal retrieval, at least one of the query or the candidate contains an image. Given a query $q_k$, the retrieval process selects the candidate $c_* \in C$ whose regional representation $c_* \otimes r_{c_*}$ is most semantically aligned with $q_k \otimes r_{q_k}$, denoted as:

$$c_* = \underset{\{c\} \in C}{\arg\max}[\langle \Phi(q_k \otimes r_{q_k}), \ \Phi(c \otimes r_c) \rangle], \tag{1}$$

where $\Phi(\cdot)$ denotes the function that embeds the query and candidates into vector representations, and $\langle \cdot, \cdot \rangle$ denotes the similarity function, such as the cosine similarity. In practice, the system usually returns a ranked list of the top-k relevant candidates instead of just the highest-scoring match.

## 3.2 MODEL DESIGN

To achieve better multimodal representations, our model LEMUR is built on vision-language models (VLMs), inspired by Liu et al. (2025). The overall framework is shown in Figure 2. First, the context encoder processes the entire image query into visual tokens, capturing holistic image content. The region-aware encoder then refines the middle-layer features of the context encoder to extract fine-grained visual tokens, which contain fine-grained information about the region of interest. Finally, the LLM backbone integrates these multimodal inputs into embeddings for retrieval.

**Region-Aware Encoder.** Building fine-grained representations for retrieval presents two main challenges. The first challenge is *to capture detailed regional features while also considering the background context to ensure accurate interpretation.* Existing methods (Jing et al., 2024; Xie et al., 2025) use RoIAlign to crop features, which leads to the loss of background information and misinterpretation of regions. On the other hand, Yu et al. (2025b) introduces excessive background information and interferes with the fine-grained features. The second challenge is *to preserve the ability to generate global representations with rich semantics.* This is crucial because models designed specifically for fine-grained retrieval should also support tasks that require background information, such as Task 4 in Figure 1b. However, even the most advanced fine-grained models inevitably lose much of the background. Directly adopting such feature extraction methods significantly reduces performance on image-level retrieval tasks.

To address these two issues, we propose the Region-Aware Encoder (RAE), as shown in Figure 2. For the first challenge, RAE extracts fine-grained features and filters out background noise using a layer-wise coordination paradigm. To be specific, firstly, the image background is encoded into

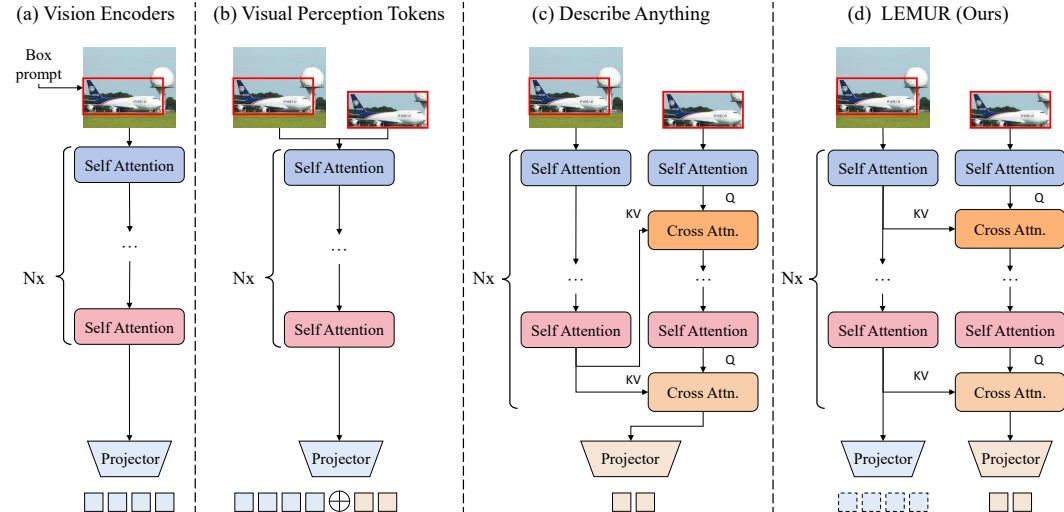

Figure 3: Comparison of vision backbone architectures and regional prompting strategies in previous methods. (a) Vanilla methods use a single ViT and overlay the query region onto the input image as a visual prompt. (b) VPT treats the region as a separate image and concatenates it with the full image. (c) DAM extracts the regional features from the last-layer's image-level visual embeddings. Both ViTs share the same projector. (d) LEMUR uses separate projectors to avoid corrupting contextual information and adopts layer-wise cross-attention to enhance fine-grained representations.

contextual vision tokens through CE; Then, for the region of interest, it is first encoded using self-attention in RAE, then refined through a cross-attention module. As shown in Figure 2, the query is from the RAE, while the key and value come from the hidden states of the corresponding layer in CE. This process can be formulated as:

$$
\begin{aligned}
h_{\text{RAE}}^{i+1} &= h_{\text{RAE}}^i + \alpha \cdot \text{xattn}\left(h_{\text{RAE}}^i, h_{\text{CE}}^i, h_{\text{CE}}^i\right), \\
h_{\text{CE}}^{i+1} &= h_{\text{CE}}^i + \text{attn}\left(h_{\text{CE}}^i, h_{\text{CE}}^i, h_{\text{CE}}^i\right),
\end{aligned}
\tag{2}
$$

where $h_{RAE}^i$ is RAE's hidden state on layer $i$, $\alpha$ is a learnable weight parameter initialized to zero in the beginning, and $xattn$ is an operator that takes arguments in the order (q, k, v). The cross-attention module is designed to extract region-specific features from CE's global representation while eliminating background noise. It offers a more comprehensive information fusion than RoIAlign for regions of all sizes. In addition, our layer-wise design capitalizes on the hierarchical features of ViTs, where shallow layers hold low-level details and deep layers contain high-level semantics. Compared with the encoder-decoder-like methods (Lian et al., 2025) in Figure 3c, which only use the encoder's final-layer hidden states, our method can capture more detailed cues from early layers, such as shape and texture.

For the second challenge, our key insight is to decouple the vision encoders. Although RAE tokens and CE tokens are both visual tokens, they are projected into distinct semantic spaces via separate projectors, allowing them to represent different levels of information. In addition, to prevent excessive semantic divergence and reduce the parameter amount, CE and RAE share parameters in self-attention modules. This simple yet effective decoupled design offers an advantage: we can train the RAE to generate highly fine-grained features by updating only its cross-attention module and projector. It leaves the image-level features from the CE tokens entirely unaffected, allowing the model to access contextual information from the CE tokens when necessary. Thus, LEMUR can improve region-level retrieval without compromising tasks that require rich contextual understanding.

**Regional Visual Prompts.** Since MLLMs do not support regional inputs, conveying regions of interest to the multimodal language model may largely impact the performance. There are several existing methods attempting to address this issue. As illustrated in Figure 3a, the most straight-forward solution (Yang et al., 2023) is using textual prompts with a set of marks over the image. Yet, this heavily relies on the VLM's visual grounding capability, which is still insufficient. This often leads to incorrect region identification and hallucination. Similarly, another strategy is to crop the region as a single image and discard the background. But the model needs contextual cues to

interpret the region correctly. For example, in Task 1 of Figure 1b, the model might mistake the umbrella for white cloth without the background of the shops. Alternatively, VPT (Yu et al., 2025b) treats regions as auxiliary images as depicted in Figure 3b. The region is concatenated after the original image and processed jointly with the same visual encoder. However, forcing the shared ViT to specialize in extracting fine-grained features can compromise its global representation ability.

To overcome the shortcomings of the above approaches, we adopt a different strategy: regional inputs are still treated as auxiliary images, but they are encoded via the separate encoder RAE, as shown in Figure 3d. Unlike the shared-projector architectures such as Figure 3c, this separation allows for enhancing RAE's fine-grained understanding ability without impacting the context encoder's parameters. Moreover, our method is flexible, allowing the localized tokens to be used either in isolation or concatenated with global representations as a complementary source of information.

**VLM For Multimodal Embedding.** Similar to prior work (Jia et al., 2021), we add a special token `[EMB]` to the end of the instruction prompt and use hidden states on this token as embeddings. The special token functions as a learnable query, enabling the model to summarize relevant information into an embedding from the knowledge space of the language model.

### 3.3    TRAINING PIPELINE

**Stage-I: RAE Pretraining.** We adopt the detailed localized captioning task (Lian et al., 2025) as the pretext task to enhance RAE's fine-grained visual understanding. This task requires the model to generate attribute-rich regional descriptions under a next-token prediction paradigm. During this stage, we keep the RAE and its connector trainable and freeze the language backbone. Given the token sequence length $T$, token $x_i$, and RAE's trainable parameter $\theta$, the loss can be formulated as:

$$\mathcal{L}_{rae} = -\frac{1}{T}\sum_{t=1}^{T}\log P(x_t \mid x_{<t};\theta).\tag{3}$$

By compelling the language model to generate extremely detailed captions solely from RAE's visual tokens, it can enhance the RAE's fine-grained understanding capability.

**Stage-II: Language-Only Contrastive Learning.** In this stage, we perform contrastive learning with InfoNCE (Oord et al., 2018) loss to efficiently transfer the LLM's language ability into embedding capabilities. We only use large-scale text-only pairs, as involving image–text pairs brings higher computational cost while yielding comparable performance (Liu et al., 2025). After this phase, the model acquires an initial capacity for multimodal retrieval.

**Stage-III: Regional Contrastive Learning.** In the final stage, we conduct instruction tuning using a mixture of global-level and region-level contrastive pairs. Unlike previous methods (Xie et al., 2025) which only align region-level images and texts, we enable the construction of negative pairs between global and regional samples to enhance the retrieval performance at both levels. We continue to optimize the language backbone with InfoNCE loss, which can be formulated as :

$$\mathcal{L}_{rcl} = -\frac{1}{N}\sum_{i=1}^{N}\log\left[\frac{\exp\left[\langle\Phi(q_i \otimes r_{q_i}), \Phi(c_i \otimes r_{c_i})\rangle/\tau\right]}{\sum_{j=1}^{N}\exp\left[\langle\Phi(q_i \otimes r_{q_i}), \Phi(c_j \otimes r_{c_j})\rangle/\tau\right]}\right],\tag{4}$$

where $\langle\cdot,\cdot\rangle$ denotes the inner product, $\tau$ is the temperature parameter, and $N$ is the batch size.

## 4    FGMB: A FINE-GRAINED MULTIMODAL RETRIEVAL BENCHMARK

We introduce FGMB, a novel benchmark specifically designed to train and evaluate models on fine-grained retrieval tasks. The benchmark includes a total of 200k contrasitive pairs for training and 25k for testing. These examples are collected from six diverse datasets and organized into two meta tasks that reflect different levels of retrieval granularity:

**Metatask 1: Region-Level Retrieval.** This metatask focuses on retrieving content based solely on the information within a region. Although contextual information may aid interpretation, it is unnecessary for successful retrieval. This metatask includes three subtasks covering three retrieval settings: **Task 1** involves image-to-text retrieval, **Task 2** handles text-to-image retrieval, and

Table 1: Comparison of methods on FGMB. $q^t$ denotes text queries, $q^i$ image queries, and $c^t$ text candidates. Results in **bold** and underlined mark the best and second-best, respectively. Notably, LEMUR-8B-zs, a zero-shot model without regional contrastive learning, already surpasses most baselines, while LEMUR-8B achieves state-of-the-art performance.

| Methods | Task 1 ($q^t \rightarrow c^i$) | | Task 2 ($q^i \rightarrow c^t$) | | Task 3 ($q^i \rightarrow c^i$) | | Task 4 ($q^i, q^t \rightarrow c^i$) | | | Avg. |
|---|---|---|---|---|---|---|---|---|---|---|
| | sam | coyo | sam | coyo | xgoods | xnote | vismin | imgdiff | xgoods | |
| | R@5 | R@5 | R@5 | R@5 | R@5 | R@5 | R@1 | R@1 | R@5 | |
| *Coarse-grained* | | | | | | | | | | |
| CLIP | 36.3 | 64.4 | 44.7 | 70.2 | 48.8 | 54.1 | 50.4 | 45.4 | 63.4 | 53.1 |
| EVA-CLIP | 48.5 | 58.7 | 48.9 | 57.4 | 50.0 | 60.7 | 53.6 | **82.2** | 66.6 | 58.5 |
| SigLIP | 54.3 | 78.5 | 51.3 | 81.6 | 74.8 | 78.4 | 65.7 | 77.3 | 82.8 | 71.6 |
| SigLIP2 | 46.3 | 74.2 | 55.9 | 74.7 | 76.9 | 79.0 | 30.5 | 51.6 | 76.7 | 62.9 |
| BLIP | 35.3 | 45.8 | 28.4 | 42.1 | 65.4 | 74.8 | 33.2 | 45.3 | 68.8 | 48.8 |
| BLIP2 | 30.6 | 38.2 | 35.1 | 48.1 | 38.6 | 54.8 | 57.2 | 68.5 | 63.9 | 48.3 |
| mmE5 | 50.4 | 55.2 | 42.2 | 51.4 | 44.4 | 59.6 | 80.9 | 75.3 | **94.4** | 61.5 |
| MM-EMBED | 42.5 | 46.4 | 35.2 | 32.1 | 35.8 | 48.7 | 82.9 | 72.5 | 87.0 | 53.7 |
| VLM2VEC | 39.8 | 49.9 | 37.3 | 48.5 | 33.7 | 43.7 | 79.3 | 72.2 | 90.1 | 54.9 |
| UniIR-BLIP$_{FF}$ | 31.4 | 39.7 | 19.9 | 32.2 | 24.2 | 36.4 | 77.0 | 52.7 | 71.1 | 42.7 |
| UniIR-CLIP$_{SF}$ | 41.0 | 49.2 | 36.4 | 43.7 | 33.7 | 46.3 | 69.5 | 78.3 | 83.8 | 53.5 |
| LamRA-Ret | 35.5 | 44.0 | 35.7 | 45.3 | 44.4 | 67.0 | 82.6 | 62.8 | 86.0 | 55.9 |
| *Fine-grained* | | | | | | | | | | |
| FG-CLIP | 48.5 | 58.7 | 48.9 | 57.4 | 50.0 | 60.7 | 53.6 | **82.2** | 66.6 | 58.5 |
| DreamLIP | 37.9 | 63.3 | 45.9 | 71.3 | 49.1 | 49.3 | 55.7 | 54.9 | 69.6 | 55.2 |
| FineCLIP | 39.4 | 72.9 | 44.2 | 71.4 | 60.5 | 67.7 | 57.2 | 28.9 | 23.7 | 51.8 |
| *Ours* | | | | | | | | | | |
| LEMUR-8B-zs | 59.5 | 80.3 | 72.4 | 80.9 | 85.1 | 86.3 | 81.1 | 70.8 | 92.8 | 78.8 |
| LEMUR-3B | 58.1 | 83.9 | 70.6 | 84.3 | 88.3 | 91.7 | 83.1 | 78.0 | 81.3 | 79.9 |
| LEMUR-8B | **69.1** | **88.9** | **86.5** | **87.3** | **92.3** | **93.6** | **83.4** | 81.1 | 93.6 | **86.2** |

**Task 3** addresses image-to-image retrieval. For Tasks 1 and 2, we utilize public datasets including COCO (Lin et al., 2014), SAM (Kirillov et al., 2023b), and FinHARD (Xie et al., 2025). Based on the annotated region of interest, we regenerate longer and more detailed captions using LLMs to increase the retrieval difficulty. For Task 3, there is a unique challenge that public datasets lack cross-scene region-level pairs of the same object. To address this, we construct datasets XGoods and XNote from E-commerce scenarios, where promotional and buyer-committed images often depict the same item in different contexts. All data are subjected to face anonymization, privacy filtering, and manual annotation at the region level to ensure high data quality and ethical integrity.

**Metatask2: Context-Level Retrieval.** This setting introduces **Task 4**, image-text-to-image retrieval, which evaluates whether models can understand not only the target region but also the surrounding contextual cues. As illustrated in Figure 3b, multiple candidates might contain the flower, but only those with a metal vase in the background are considered correct. We leverage datasets like VisMin (Awal et al., 2024) and ImgDiff (Jiao et al., 2024), which naturally include positive and negative region-level image pairs created via image editing. Furthermore, we sample a subset from XGoods and automatically annotate it with LLMs as a held-out split for evaluation.

More detailed dataset statistics and preprocessing methods are provided in Appendix A.

## 5 EXPERIMENTS

### 5.1 IMPLEMENTATION DETAILS

We adopt Qwen2-VL (Wang et al., 2024) models with 3B and 8B parameters as the backbone and train them using the three-stage strategy outlined in Section 3.3. In Stage 1, we pretrain the Region-Aware Encoder using 800k region-level image-text pairs from the DAM (Lian et al., 2025) and PAM (Lin et al., 2025) datasets. During this stage, we only update the parameters of the RAE's cross-attention layers and its projector. In stage 2, we use the datasets of Natural Language Inference (NLI) (Gao et al., 2021), HotpotQA (Yang et al., 2018), and MSMARCO (Nguyen et al., 2016), totalling approximately 780k text pairs. In stage 3, we further fine-tune using mixed data that includes 1.8M image-level pairs from M-BEIR (Wei et al., 2024) and 200k region-level fine-grained pairs from FGMB. More training details can be found in Appendix B.

Table 2: Performance on the image-level multimodal retrieval benchmarks from M-BEIR. Although specifically designed for fine-grained retrieval, LEMUR achieves comparable or even better results than strong baselines on image-level retrieval tasks.

| Methods | $q^t \to c^i$ | | $q^t \to (c^i, c^t)$ | | $q^i \to c^t$ | | | $q^i \to c^i$ | $(q^i, q^t) \to c^t$ | | $(q^i, q^t) \to c^i$ | $(q^i, q^t) \to (c^i, c^t)$ | Avg. |
|---|---|---|---|---|---|---|---|---|---|---|---|---|---|
| | COCO | F200K | EDIS | WebQA | COCO | F200K | NIGHTS | InfoSeek | F200IQ | CIRR | OVEN | InfoSeek | |
| | R@5 | R@10 | R@5 | R@5 | R@5 | R@10 | R@5 | R@5 | R@10 | R@5 | R@5 | R@5 | |
| *CLIP-based methods* | | | | | | | | | | | | | |
| CLIP | 61.1 | 6.6 | 43.3 | 45.1 | 79.0 | 7.7 | 26.1 | 20.5 | 7.0 | 13.2 | 38.8 | 26.4 | 31.2 |
| SigLIP | 75.7 | **36.5** | 27.0 | 43.5 | 88.2 | **34.2** | 28.9 | 25.1 | 14.4 | 22.7 | 41.7 | 27.4 | 38.8 |
| BLIP | 74.4 | 15.9 | 26.8 | 20.3 | 83.2 | 19.9 | 27.4 | 10.2 | 2.3 | 10.6 | 27.4 | 16.6 | 27.9 |
| BLIP2 | 63.8 | 14.0 | 26.9 | 24.5 | 80.0 | 14.2 | 25.4 | 5.5 | 4.4 | 11.8 | 27.3 | 15.8 | 26.1 |
| UniIR-BLIP$_{FF}$ | 79.7 | 26.1 | 50.9 | 79.8 | 89.9 | | **33.0** | 22.4 | 29.2 | 52.2 | 55.8 | 33.0 | 48.4 |
| UniIR-CLIP$_{SF}$ | 81.1 | 18.0 | 59.4 | 78.7 | **92.3** | 18.3 | 32.0 | 27.9 | 24.4 | 44.6 | 67.6 | 48.9 | 49.4 |
| *VLM-based methods* | | | | | | | | | | | | | |
| Qwen2-VL-7B | 55.1 | 5.0 | 26.2 | 9.4 | 46.6 | 4.0 | 21.3 | 22.5 | 4.3 | 16.3 | 43.6 | 36.2 | 22.3 |
| LamRA-Ret | 81.5 | 28.7 | 62.6 | 81.2 | 90.6 | 30.4 | 32.1 | 52.1 | **33.2** | 53.1 | 76.2 | 63.3 | 57.1 |
| *Ours* | | | | | | | | | | | | | |
| LEMUR-8B | **81.6** | 28.9 | **62.7** | **81.8** | 90.3 | 30.6 | 31.6 | **52.7** | **33.2** | 53.3 | 76.4 | **64.5** | 57.3 |

## 5.2 BASELINES AND BENCHMARKS

We compare our model with both image-level and fine-grained retrieval models, including CLIP (Radford et al., 2021), SigLIP (Zhai et al., 2023), SigLIP2 (Tschannen et al., 2025), BLIP (Li et al., 2022), BLIP2 (Li et al., 2023), UniIR-CLIP$_{SF}$ (Wei et al., 2024), UniIR-BLIP$_{FF}$ (Wei et al., 2024), MME5 (Chen et al., 2025b), VLM2VEC (Jiang et al., 2024b), MM-Embed (Lin et al., 2024), FG-CLIP (Xie et al., 2025), FineCLIP (Lin et al., 2023), DreamLIP (Zheng et al., 2024), and LamRA-Ret (Liu et al., 2025). To standardize the evaluation, we use the image with bounding boxes drawn as visual prompts for all baselines. For methods that support regional prompts (e.g., FG-CLIP), we report the best result of the two prompt methods. To evaluate LEMUR's zero-shot ability, we further provide LEMUR-8B-zs, which has not been trained on any regional contrastive pairs. We evaluate our method on FGMB. For most tasks, we report recall@5. For vismin and imgdiff, where recall@5 is often saturated, we report recall@1 instead. We also evaluate its image-level retrieval performance on some splits from M-BEIR. Following the common practice as Wei et al. (2024), we report recall@5 for most subtasks and recall@10 on F200K (Wu et al., 2021).

## 5.3 ANALYSIS

Our main results are shown in Table 1. Across all modality combinations and both meta tasks, our method consistently outperforms existing baselines. Several key observations can be drawn from the results: (1) Even without task-specific fine-tuning, the zero-shot performance of our models already exceeds the best baseline SigLIP by 7.2 points, highlighting the robustness of our RAE features. (2) After training with FGMB data, the 3B model already surpasses some 11B models (e.g., vlm2vec), and the 8B version achieves an extra average improvement of 7.4 points compared to the zero-shot model, demonstrating the effectiveness of our data and training strategy. (3) Models that perform well in Tasks 1–3 (e.g., SigLIP) generally underperform in Task 4, and vice versa (e.g., MME5). This reveals the inherent tension between local and global representation learning. The models focusing on region-level features may lose the comprehensive perception of the contextual cues. (4) Although the 8B version outperforms most baselines, FG-CLIP and MME5 exceed it by 0.95 points on average in task 4. This can be partially attributed to the task's higher dependency on global background context, where fine-grained features provided by LEMUR are less helpful. Meanwhile, these models are trained with millions of synthetic data. LEMUR achieves close performance with only 200k regional examples, reflecting the high data efficiency and strong generalization abilities.

LEMUR's image-level retrieval performance is reported in Table 2. We observed performance boosts on the majority of splits. The improvement is particularly notable on complex tasks like InfoSeek (image-text to image-text), where it increases by 1.2 points. This indicates that our regional contrastive learning not only preserves but even enhances global retrieval performance.

Table 3: Ablations on the RAE's architecture using LEMUR-8B.

| xattn | separate proj. | layer-wise | Task1 | Task2 | Task3 | Task4 | FGMB Avg. | M-BEIR Avg. |
|-------|----------------|------------|-------|-------|-------|-------|-----------|-------------|
| ✗ | ✗ | ✗ | 48.1 | 52.5 | 70.4 | 75.2 | 63.1 | 57.1 |
| ✓ | ✗ | ✗ | 77.2 | 83.5 | 51.7 | 52.2 | 66.0 | 27.2 |
| ✓ | ✓ | ✗ | 76.6 | 83.8 | 89.4 | 81.3 | 82.6 | 57.1 |
| ✓ | ✓ | ✓ | **79.0** | **86.9** | **92.9** | **86.0** | **86.2** | **57.3** |

Table 4: Ablations on the training data. The combination of the regional data from FGMB, image-level data from MBEIR, and the pure-text contrastive pairs achieves the best performance on FGMB.

| Model | Language | Image-level | Regional | Task1 | Task2 | Task3 | Task4 | Avg. |
|-------|----------|-------------|----------|-------|-------|-------|-------|------|
| LEMUR-3B | ✓ | | | 61.3 | 65.4 | 74.5 | 71.1 | 68.4 |
| | ✓ | ✓ | | 65.1 | 62.4 | 64.6 | 80.1 | 69.4 |
| | ✓ | ✓ | ✓ | **71.0** | **77.4** | **90.0** | **80.8** | **79.9** |
| LEMUR-8B | ✓ | ✓ | ✓ | 79.0 | 86.9 | 92.9 | 86.0 | 86.2 |

# 6 ABLATIONS

**ViT Architecture.** To validate the effectiveness of RAE's architecture, we conduct ablation studies using different architectural variants in Table 3. When using only the naive ViT backbone without cross-attention, the model exhibits reasonable global representation capability on M-BEIR, but remains suboptimal in the fine-grained benchmark. Relying solely on xattn compromises the performance on both benchmarks due to the trade-off between local and global information. By using separate projectors, we can balance this trade-off. Furthermore, introducing the layer-wise coordination improves semantic alignment between CE and RAE, yielding additional performance gains.

**Regional Prompt Strategy.** To demonstrate that LEMUR's improvements primarily stem from the model and data rather than the prompt strategy, we compare the zero-shot version with LamRA-Ret, a state-of-the-art retrieval model trained with the same dataset. We evaluate three prompt methods as mentioned in Section 3.2: (1) Crop the target region and discard the background; (2) Draw the region on the image as a visual marker; and (3) Concatenate the cropped region as an auxiliary image. However, even with the best prompting strategy, LamRA-Ret still falls short of LEMUR, indicating that our model's advantage arises from its architecture rather than the prompting strategy.

Table 5: Comparison of LEMUR and MLLMs under different prompting strategies on FGMB. The first three rows are based on LamRA-Ret. Naive prompt-based adaptation of image-level retrieval models to fine-grained retrieval is suboptimal.

| Prompt | T1 | T2 | T3 | T4 | Avg. |
|--------|-----|-----|-----|-----|------|
| Crop | 55.3 | 28.3 | 79.2 | 59.2 | 55.9 |
| Draw box | 39.8 | 40.5 | 55.7 | 77.1 | 55.9 |
| Auxiliary image | 48.1 | 52.5 | 70.4 | 75.2 | 63.1 |
| LEMUR-8B-zs | **69.9** | **76.7** | **85.7** | **81.6** | **78.8** |

**Training Data.** Table 4 presents ablations on different training data sources. We compare the use of language-only data from Stage 2, image-level data from M-BEIR, and regional-level contrastive pairs from FGMB. While adding M-BEIR's data provides a modest improvement, we observe a notable boost in fine-grained retrieval performance after integrating region-level contrastive pairs from FGMB. This reveals the effectiveness of our data.

# 7 CONCLUSION

We present LEMUR, a novel framework designed for diverse fine-grained multimodal retrieval tasks. Equipped with the Region-Aware Encoder, it significantly enhances the region-level representation capability. We further build the FGMB benchmark to enable the training and evaluation of fine-grained retrieval models. Compared to previous approaches, LEMUR sets a new state-of-the-art in performance. Future work will explore extending LEMUR's ability to visual document retrieval and integrating it into retrieval-augmented generation systems.

**Usage of LLM.** In this work, we employed a Large Language Model (LLM) to polish the prose and enhance the overall quality of the text.

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

# APPENDICES

## A   DATASET DETAILS

This section details the construction of FGMB. The benchmark is aggregated from a total of six data sources. These include four publicly available datasets-SAM, COYO, ImgDiff, and Vismin-and two proprietary datasets: XGoods and XNote, which we collected from an e-commerce platform to cover a broader range of fused-modal scenarios. Using the bounding box annotations from these datasets, we created regional query-candidate pairs. To guarantee that the benchmark is sufficiently challenging and discriminative, each query is paired with one to six hard negatives across all tasks.

**SAM.** This dataset is instrumental for building Tasks 1 and 2. We first converted the segmentation masks from SAM into bounding boxes by taking the maximum coordinates along each axis. Each region was then paired with a corresponding long caption from the DAM Lian et al. (2025) training set. To ensure an adequate supply of hard negatives, we required that each candidate pool include at least two distinct regions. For both Task 1 and Task 2, this process yielded 30k samples for training and 1.5k for evaluation.

**COYO.** From the COYO dataset, we utilized bounding boxes previously generated by FG-CLIP and subsequently re-captioned each region with DAM Lian et al. (2025). Employing much longer captions allows for a richer alignment between the image region and its associated semantic concepts. These region-caption pairs naturally form the basis for text-to-image and image-to-text retrieval tasks, for which we prepared 1.5k evaluation samples each.

**ImgDiff.** As an image editing dataset, each sample in ImgDiff contains a source image, an editing instruction, an edited image, and a bounding box indicating the modified area. This structure lends itself naturally to an image-text-to-image retrieval task, where the query is a composite of the source image and the instruction, and the positive candidate contains the target region. When constructing the test split, we exclusively retained samples with two or more editing instructions to serve as hard negatives. This split comprises 21k training samples and 1.3k test samples.

**Vismin.** The structure of Vismin is analogous to that of ImgDiff; the primary difference is that Vismin contains real-world photographs, whereas ImgDiff's images are in the AI-generated style. We applied the same processing approach: the source image and edit instruction form the query, and the modified image serves as the positive candidate. From this dataset, we selected 46k samples for training and 1.3k queries for testing.

**XGoods.** Recognizing the limited availability of public datasets for region-level image-to-image retrieval, we collected XGoods from an e-commerce platform. Unlike datasets such as NIGHTS, which only contain the same object in similar backgrounds, XGoods offers more complex and realistic scenes. In this task, a commercial product image acts as the query, while a user-uploaded photo of the same item serves as the positive candidate. The bounding boxes of the query object were initially annotated automatically and then manually refined. We paired each query with six highly relevant products as hard negatives. For Task 3, we allocated 69k samples for training and 3.3k for evaluation. For Task 4, we also built a 3.4k data test split as a held-out dataset.

**XNote.** Similar to XGoods, XNote was also collected from social media content on the same e-commerce platform. The main difference is its focus on images from users' daily lives rather than commercial product displays. The data was processed using the same procedure as XGoods, from which we carefully selected 4.2k query-candidate pairs for the final test set.

## B   TRAINING DETAILS

### B.1   TRAINING PARAMETERS

In the first stage, we use the batch size 64 and the learning rate of $1 \times 10^{-4}$. Only RAE's cross-attention modules and the projector are trainable. In the second stage, the batch size is 1104 and the learning rate is $1.1 \times 10^{-4}$. We only update the language model backbone's parameters for one epoch using pure-text contrastive pairs. In the final stage, we fine-tune the language model backbone for two epochs on multimodal contrastive pairs, with a learning rate of $2 \times 10^{-4}$ and a batch size of 1104. To save the GPU memory for a larger batch size, which is crucial in contrastive learning,

we use LoRA to fine-tune the LLM. The LoRA rank is set to be 64 and 128 for stage two and stage three, respectively. All three stages can be completed within 20 hours on 24 NVIDIA H20 GPUs.

## B.2 TRAINING STRATEGY ABLATIONS

During the training stage three, we develop the **mixed visual prompting strategy** to balance contextual and regional information. In particular, for tasks 1 to 3 in the FGMB training split, which rely more on local details, we provide only RAE tokens to the LLM. For task 4, which requires contextual understanding, we concatenate CE tokens with RAE tokens. This design allows the model to leverage CE tokens for context without injecting excessive background signals into RAE, thereby preserving fine-grained representation quality.

We compare the mixed visual prompting strategy with some approaches in 6. This strategy proves to be able to provide consistent performance improvements across both meta tasks. As shown in Table 6, we compare several alternative training strategies. The "crop" setting uses only the RAE tokens for both meta tasks; the "concat" setting uses the concatenation of context encoder (CE) and RAE tokens as visual input; and the "random" strategy randomly selects either "crop" or "concat" with a 50% probability for each meta task. We observe that our strategy achieves the best performance. This is mainly because of the trade-off between the two metatasks. Specifically, metatask 2 requires contextual information for comprehensive representation, while metatask 1 demands a highly localized, fine-grained understanding. Excessive context may thus act as a distraction in the latter. Therefore, we provide the CE tokens when training on metatask2, so that the model can access the global-level representations. Meanwhile, we use only RAE tokens on metatask 1 to prevent introducing background noise.

Table 6: Ablations on the training strategy in Stage-III using 3B models, evaluated on FGMB.

| Strategy | Task1 | Task2 | Task3 | Task4 | Avg. |
|---|---|---|---|---|---|
| crop | 68.2 | 78.4 | 89.3 | 77.9 | 78.4 |
| concat | 68.3 | 77.3 | 88.5 | 71.4 | 75.8 |
| random | 67.8 | 78.9 | 89.2 | 73.0 | 76.7 |
| mixed visual prompts | 71.0 | 77.4 | 90.0 | 80.8 | 79.9 |

# C ETHICS AND REPRODUCIBILITY STATEMENTS

## C.1 ETHICS STATEMENT

**Dataset and Licensing:** Our dataset, FGMB, consists of two parts: one sourced from publicly available datasets and the other from private data collected from e-commerce platforms. We are authorized by the company to obtain and share the data. The dataset is released under CC BYNC-SA 4.0 license and can be used for noncommercial purposes. The private data has undergone privacy filtering to ensure compliance with data protection regulations. Ultimately, the entire dataset will be open-sourced for public use.

The authors declare no competing interests or conflicts of interest related to this work.

## C.2 REPRODUCIBILITY STATEMENT

We are committed to ensuring the reproducibility of our work. To facilitate this, we provide detailed descriptions of our methodology, data, and experimental setup throughout the paper and its appendices. We plan to release the code for LEMUR, our pre-trained models (LEMUR-3B, LEMUR-8B), and the FGMB benchmark upon publication of this paper.

