# OpenReview forum: "LEMUR: Leveraging Vision-Language Models for Fine-Grained Multimodal Retrieval"
_ICLR.cc/2026/Conference — ICLR 2026 Conference Withdrawn Submission_

### Official Review · Reviewer_417X · 2025-10-25

**Soundness:** 2
**Presentation:** 2
**Contribution:** 2
**Rating:** 2
**Confidence:** 4

**Summary:**

The authors introduce LEMUR, a V-L framework for fine-grained multi-modal retrieval. It features a Region-Aware Encoder (RAE) that integrates multi-layer cross-attention with a global Context Encoder (CE), to preserve both local and global contexts. The authors propose a three-stage training pipeline. They also introduce FGMB, a large fine-grained benchmark.

**Strengths:**

- Clear motivation: the authors describe their architectural motivation for decoupling regional vs global signals.
- The paper present comprehensive comparisons with many previous methods for retrieval.

**Weaknesses:**

## Related Work ##
While the paper’s core focus is retrieval, its discussion of related work is incomplete and somewhat misleading. The authors frame their approach mainly in the context of CLIP-based retrieval, overlooking the long line of pre-CLIP and concurrent retrieval research that used unimodal or other multimodal encoders [1, 2]. Text–image retrieval is a well-established field, and the paper should not imply that CLIP introduced the task itself. Similarly, the task referred to as “MetaTask2” corresponds to Composed Image Retrieval in prior literature [3]; this and other sub-tasks deserve proper acknowledgment and discussion. A more comprehensive review and citation of prior retrieval studies would better situate LEMUR’s contributions and clarify how it advances the existing body of work.

## Evaluation ##
Although the paper presents comparisons with numerous baselines, the evaluation remains insufficient to convincingly demonstrate the method’s superiority:
- A) The main comparison (Table 1) relies entirely on the authors’ own benchmark (FGMB). While LEMUR achieves strong results there, performance on a self-constructed benchmark is inherently less persuasive. It is important to include results on established public benchmarks for each retrieval sub-task. For instance, using existing datasets for Composed Image Retrieval [2].
- B) The comparison in Table 2 appears unfair: prior models were not necessarily trained or fine-tuned on the authors’ chosen benchmarks and are therefore evaluated in a zero-shot setting, whereas LEMUR is trained on them. This discrepancy skews the results. For example, BLIP-2’s reported Recall@5 on COCO T2I (63.8%) is much lower than its original performance reported in their paper [4] (86.5-87.7%, depending on backbone size).
- C) The paper also omits a key baseline: fine-tuning or evaluating the underlying pre-trained backbone (Qwen2-VL) without the proposed architectural modifications (as described in lines 281-284). Without this comparison, it remains unclear how much of the reported improvement stems from LEMUR’s design versus the backbone’s inherent strength. Table 3 may relate to this, but its setup is not explained clearly in the text.

## Paper Writing ##
I appreciate the authors’ contributions and the technical depth of the work; however, the paper does not yet feel ready for publication in its current form. Several writing and organization issues significantly affect readability and completeness:

- Certain sections should be reorganized for logical flow. For instance, Figure 2 is referenced before Figure 1, which disrupts the narrative order.
- The Introduction and Related Work sections need to more clearly position the addressed retrieval tasks within the terminology and context of the main literature (see the first comment on related work).
- Tables 3, 4, and 5 are insufficiently discussed and omit essential details. For example, the meaning of the “xattn” symbols in Table 3 is unclear - does it correspond to the cross-attention component in Eq. 2, and do “X”/“V” denote $\alpha=0$ and $\alpha=1$ respectively, or something else? Similarly, Table 5 is not referenced anywhere in the text, and the definitions of T1/T2/T3/T4 are missing.
- Finally, the paper lacks a limitations or discussion section. No weaknesses or failure cases of LEMUR are analyzed, which would be important for a balanced and transparent presentation.


[1] Li, K., Y. Zhang, K. Li, et al. Visual semantic reasoning for image-text matching. In ICCV,
pages 4653–4661. 2019

[2] Li, J., Selvaraju, R., Gotmare, A., Joty, S., Xiong, C. and Hoi, S.C.H., 2021. Align before fuse: Vision and language representation learning with momentum distillation. Advances in neural information processing systems, 34, pp.9694-9705

[3] Song, X., Lin, H., Wen, H., Hou, B., Xu, M. and Nie, L., 2025. A comprehensive survey on composed image retrieval. ACM Transactions on Information Systems.

[4] Li, J., Li, D., Savarese, S. and Hoi, S., 2023, July. Blip-2: Bootstrapping language-image pre-training with frozen image encoders and large language models. In International conference on machine learning (pp. 19730-19742). PMLR.

**Questions:**

See "Weaknesses" above

---

### Official Review · Reviewer_mxwK · 2025-10-30

**Soundness:** 3
**Presentation:** 4
**Contribution:** 3
**Rating:** 6
**Confidence:** 3

**Summary:**

The paper is about LEMUR that is a VLM based fine grained retrieval framework that enhances regional representations without compromising image level retrieval performance. The core contribution is the addition of a Region Aware Encoder that extracts detailed features from query regions to complement global image representation. Additionally, the manuscript introduce a new benchmark named FGMB of 225k region level contrastive pairs covering two metatasks and four multimodal retrieval scenarios.

**Strengths:**

+ The region aware encoder (RAE) appears to be innovative as it addresses the fundamental tension between local and global representation learning through decoupled projectors. The results reported in Table 1 shows significant improvement compared to the baselines and both coarse and fine grained approaches.
+ The motivation of the RAE componenet is clear to encourage detailed regional representations, and the following pipeline is similarly well motivated. The language only contrastive learning converts language generation capability into embedding capability and the regional contrastive learning addresses fine grained retrieval performance.
+ The paper is relatively easy to read and well presented.
+ The FGMB dataset is valuable and interesting.

**Weaknesses:**

- The paper mentions using LLMs to regenerate captions for increased difficulty, but doesn't discuss potential biases introduced by this synthetic data generation. I did not found which LLM, and details about how this was done.
- While the paper introduces the FGMB dataset, there is brief mention of why existing benchmarks were insufficient.  There are concurrent development of similar benchmarks with regions retrieval, with differences, but still related (e.g. with reasoning, visual commonsense reasoning dataset). It is also missing comparison with existing evaluation protocols (like from GPT4RoI (ECCV2024), KOSMOS-2 (arXiv:2306.14824) and others).
- There are missing related works research lines and comparisons with the proposed approach. They are similar in terms of tasks, for instance VisionLLM (Neurips2023), GPT4RoI (ECCV2024)  and ASM (ICLR2023) utilize spatial boxes with ROI-aligned features to align the region-level features into LLM word embedding space.

**Questions:**

- What's the performance impact of using different vision backbone architectures? What about LLMs to generate the synthetic data?
- How does LEMUR perform on tasks requiring multiple regions of interest simultaneously?
- How sensitive is the model to bounding box quality/precision?

---

### Official Review · Reviewer_7tUb · 2025-10-31

**Soundness:** 3
**Presentation:** 2
**Contribution:** 2
**Rating:** 2
**Confidence:** 4

**Summary:**

The paper proposes LEMUR, a VLM for fine-grained multimodal retrieval that enhances region-level understanding via a Region-Aware Encoder (RAE), localized captioning, and regional contrastive learning. The main model contribution, the RAE processes the global image and the region crop separately with cross-attention layers to keep both context and fine-grained features before feeding into the LLM. The paper also introduces the FGMB benchmark with 225K diverse contrastive pairs, including region aligned pairs. LEMUR improves on previous baseline specifically in fine-grained retrieval.

**Strengths:**

- The paper tackles an important problem of fine-grained image retrieval with region-level alignment between text and images.
- The proposed benchmark can serves for future development and evaluation of fine-grained retrieval tasks.
- The LEMUR is simple and technically sound. The RAE, although not the most efficient solution, is reasonable.

**Weaknesses:**

- The proposed tasks make sense for evaluation purposes, but their real-world applicability is questionable.
  * The region is not only given for the query, but also for every candidate. This assumption seems unrealistic. In a practical open-set retrieval setting, one likely does not have bounding boxes for each image. Even if one would automatically generate candidate bounding boxes for each image, the error rate of the object detector becomes a bottleneck. Worse, the proposed method is highly inefficient if multiple bounding boxes for the same image were to be considered.
  * In the dataset, only one bounding box exists per image (to my understanding). One could argue such a setting actually makes the task easier since the bounding box supervises which part of the image is relevant for the retrieval task (the rest can be ignored as a relevant caption does not exist). So why do competing methods show a weak performance? Likely because they are not trained to make use of this information.
  * A more realistic setting does not have bounding boxes on the candidate set and could identify the relevant region of the image.

- Some choices of the LEMUR method could be better justified and explained.
  * The alternative architectures presented in Fig. 3 are reasonable candidates. It is not clear if the ablations from Tab. 4 directly correspond to some of these versions. Most importantly, Fig 3b does not seem to be ablated which is likely the most promising alternative due to its architectural simplicity and fewer number of parameters.
  * Training details of the zero-shot variant are not clear. What is the exact difference to the full model? Are only the 200k FGMB samples omitted from training? How is RAE trained without any region (bounding box) data?
  * Compared to most other baselines, the proposed method likely is much more computationally costly due processing two images instead of one. A runtime analysis would be useful to put this into perspective.

- Novelty is limited.
  * As mentioned above, the model is specifically tuned for the use-case of having bounding boxes in the candidate set. Hence, it is not surprising that it performs well on the proposed benchmark.
  * The architectural changes in LEMUR are not too innovative and it mostly introduces more parameters and additional tokens (compute) to solve the problem.
  * The proposed dataset is mostly a collection of existing datasets. The novel additions XGoods and XNote are barely described.

- Experimental evaluation could be improved.
  * The evaluations mix too many settings without transparently disclosing fair comparisons. Tab. 2 should clearly indicate which models are trained on M-BEIR.
  * For better comparison the tables should also specify the size of each model.
  * Both Tab. 1 and 2 compare only to LamRA-ret and not the better LamRA. LEMUR performs worse on average than LamRA on M-BEIR.
  * It is not clear why Tab. 2 contains significantly fewer models than Tab. 1. Moreover, there are relevant baselines that are mentioned in the related works and have models is available, but are not evaluated. For instance, E5-V, MoCa, FLAIR.
  * In ablation Tab. 3, it is not clear how xattn can be trained without separate projectors.

**Questions:**

- How do the results change when we do not assume to have bounding box information for the candidate set?
- Could you clarify the ablation study with respect to Fig. 3? How would to comparison to 3b look like everything else being equal?
- Could you please clarify the differences of the zero-shot variant?
- What is the runtime of LEMUR compared to other baselines?
- How do other recent methods (e.g., MoCa, FLAIR) compare to LEMUR?

I am willing to increase my score if my concerns are well addressed.

---

### Official Review · Reviewer_vJCV · 2025-11-01

**Soundness:** 2
**Presentation:** 2
**Contribution:** 2
**Rating:** 2
**Confidence:** 4

**Summary:**

This paper presents LEMUR, a VLM based multimodal retrieval framework to improve fine-grained and region-level multimodal retrieval. Specifically, the paper proposes a region-aware encoder (RAE) that mirrors the original vision encoder (CE) in MLLMs. The RAE receives context information from the CE through cross-attention layers. The decoupled nature of the RAE from the CE allows the RAE to be more focused on region level visual cues. The paper also presents a new benchmark, FGMB, to evaluate fine-grained retrieval tasks.

**Strengths:**

- The problem of fine-grained multimodal retrieval is a common issue tackled in industry. The paper identifies the issue that most systems (often based on CLIP-like models) fail to perform retrieval in fine-grained tasks.
- The RAE architecture is an intuitive design with clear motivations: use the original vision encoder (or Context Encoder/CE) to extract more global information, while the cropped region is fed into the RAE. The CE feeds context information to the RAE through cross-attention layers.
- Various experiments are conducted and LEMUR is compared with other well-known models, such as CLIP, BLIP, SigLIP, and Qwen-VL.

**Weaknesses:**

- There are many statements in the paper that aren't well supported or grounded. This raises some concerns. For example, in L207, authors claim that "Yu et al. (2025b) introduces excessive background information and interferes with fine-grained features". However, I am familiar with the cited paper, and I do not understand why this statement made by the authors is true. What the paper proposes is a region-selection or re-encoding token, two tokens which serve distinct purposes, and in the case of region selection token (which is what is used for further fine-grained analysis) the background is cropped and fed into the model, so I don't understand why there is criticism for "introducing excessive background". Another example is that across Section 3.2, the authors claim the effectiveness of the RAE module; however, I feel that there is not enough analysis surrounding what the RAE focuses on (in the image) and how effectively it complements the CE (other than quantitative results in experimental resutls). Thus, it is difficult to determine whether improved performance is truly due to the architectural/fine-grained nature of the RAE module, or simply because it is specifically trained on extracted regions of the image.
- The assumption that target region (i.e., bounding box) is always present may be a bit prohibitive.
- There is no proper consideration for compute efficiency or latency. In real-world systems, especially for retrieval and search, this is a crucial aspect that needs to be considered. Does the increased latency/computation justify the performance improvement?
- Section 3.3 can be elaborated, as there is some ambiguity in how specifically retrieval is conducted. For example, to someone first reading the paper or new to the field in general, it may not be entirely clear how single feature embeddings are generated from a decoder-only language model. Furthermore, there is also ambiguity on what the prompt would be for the candidates.
- Last but not least, the idea of decoupling the encoder for more fine-grained features is not entirely novel. While this itself is fine, I would expect more insights or thorough analysis on why such design is much more effective, rather than intuitive but not-well-grounded statements. Thus, overall, I do not feel as though I have learned something new.

**Questions:**

I would appreciate if the authors addressed the concerns raised in Weaknesses.

---

### Note · Authors · 2025-11-14

**Comment:**

Thanks for the reviewers' comments. In fact, the main concerns have already been addressed in the experiments, but it seems the reviewers may not have fully understood this due to the writing and presentation. We will refine the manuscript to make the explanations clearer and resubmit it.

**Withdrawal Confirmation:**

I have read and agree with the venue's withdrawal policy on behalf of myself and my co-authors.